# SHAPLEY VALUE APPROXIMATION BASED ON k-ADDITIVE GAMES

## ABSTRACT

The Shapley value is the prevalent solution for fair division problems in which a payout is to be divided among multiple agents. By adopting a game-theoretic view, the idea of fair division and the Shapley value can also be used in machine learning to quantify the individual contribution of features or data points to the performance of a predictive model. Despite its popularity and axiomatic justification, the Shapley value suffers from a computational complexity that scales exponentially with the number of entities involved, and hence requires approximation methods for its reliable estimation. In this paper, we propose $SVAk_{\mathrm{ADD}}$, a novel approximation method that fits a $k$-additive surrogate game. By taking advantage of the assumption of $k$-additivity, we are able to compute the exact Shapley values of the surrogate game in polynomial time, and then use these values as estimates for the original fair division problem. The efficacy of our method is evaluated empirically and compared to competing methods.

## 1 INTRODUCTION

The continuous advances in computing hardware, providing cheaper and more computational power to the public, contributed to the rapid and certainly significant increase in complexity that machine learning models have experienced over the last decade. Coupled with the availability of large data sources, these complex models exhibit noteworthy predictive performances and capabilities leading to subfields such as generative AI (Gozalo-Brizuela & Garrido-Merchan, 2023). On the contrary, this development comes with an ever-rising burden to understand a model's decision-making, reaching a point at which the inner workings are beyond human comprehension, and fittingly coining the term 'black box model'. Meanwhile, societal and political influences led to a growing demand for trustworthy AI (Li et al., 2023). The field of Explainable AI (XAI) emerges to counteract these consequences, aiming to bring back understanding to the human user and developer. Among the various explanation types (Molnar, 2021), post-hoc additive explanations convince with an intuitive appeal: an observed numerical effect caused by the behavior of the black box model is divided among participating entities. This allows to interpret each assigned share to an entity as its contribution towards the behavior, e.g., the performance of a classifier (Covert et al., 2020). Beyond explainability, this allows in feature engineering to conduct feature selection by removing features with irrelevant or even harmful contributions (Cohen et al., 2005). Most popular (Marcílio & Eler, 2020) are additive feature explanations which decompose a predicted value for a particular datapoint or generalization performance on a test set among the involved features, enabling feature importance scores.

Treating this decomposition as a fair division problem opens the door to game theory which views the features as cooperating agents, forming groups called coalitions to achieve a task and collect a common reward that is to be shared. Such scenarios are captured by the simple but expressive and thus widely applicable notion of cooperative games (Peleg & Sudhölter, 2007), modeling the agents as a set of players $N$ and assuming that a real-valued worth $\nu(A)$ can be assigned to each coalition $A \subseteq N$ by a value function $\nu$. Among multiple propositions the Shapley value (Shapley, 1953) prevailed as the most favored solution to the fair division problem. The Shapley value assigns to each player a share of the collective benefit, more precisely a weighted average of all its marginal contributions, i.e., the increase in collective benefit a player causes when joining a coalition. Its popularity is rooted in the fact that it is provably the only solution concept to fulfill certain desirable axioms (Shapley, 1953) which arguably formalize and capture a widespread understanding of fairness. For example, in the context of supply chain cooperation (Fiestras-Janeiro et al., 2011), the

gain when joining a coalition and reducing costs may be shared among the companies based on the Shapley values. The greater a company's marginal contributions to the cost reduction, the greater the benefit, measured by the Shapley value, that this company should receive.

The range of domains to which the Shapley value is applicable to exceeds by far the sphere of economics as its utility has been recognized by researchers of various disciplines. Most prominently, it has recently found its way into the branch of machine learning, especially as a model-agnostic approach, quantifying the importance of entities such as features, datapoints, and even model components like neurons in networks or base learners in ensembles (see (Rozemberczki et al., 2022) for an overview). Adopting the game-theoretic view, these entities are understood as players which cause a certain numerical outcome of interest. Shaping the measure of a coalition's worth adequately is pivotal to the informativeness of the importance scores obtained by the Shapley values. For example, considering a model's generalization performance on a test dataset restricted to the feature subset given by a coalition yields global feature importance scores (Pfannschmidt et al., 2016; Covert et al., 2020). Conversely, local feature attribution scores are obtained by splitting the model's prediction value for a fixed datapoint (Lundberg & Lee, 2017). The Shapley value's purpose is not limited to provide additive explanations since it has also been proposed to perform data valuation (Ghorbani & Zou, 2019), feature selection (Cohen et al., 2007), ensemble construction (Rozemberczki & Sarkar, 2021), and the pruning of neural networks (Ghorbani & Zou, 2020). Moreover, it has been applied to extract feature importance scores in several recent practical applications, such as in risk management (Nimmy et al., 2023), energy management (Cai et al., 2023), sensor array (re)design (Pelegrina et al., 2023b) and power distribution systems (Ebrahimi & Rastegar, 2024).

The uniqueness of the Shapley value comes at a price that poses an inherent drawback to practitioners: its computation scales exponentially with the number of players taking part in the cooperative game. Consequently, it becomes due to NP-hardness Deng & Papadimitriou (1994) quickly infeasible for increasing feature numbers or even a few datapoints, especially when complex models are in use whose evaluation is highly resource consuming. As a viable remedy it is common practice to approximate the Shapley value while providing reliably precise estimates is crucial to obtain meaningful importance scores. On this background, the recently sharp increase in attention that XAI attracted, has rapidly fueled the research on approximation algorithms, leading to a diverse landscape of approaches (see (Chen et al., 2023) for an overview related to feature attribution).

**Contribution.** We contribute to the research branch of approximating the Shapley value by proposing with $SVAk_{\text{ADD}}$ (Shapley Value Approximation under $k$-additivity) a novel method based on the concept of $k$-additive games that restricts the value function to a parameterizable structure. Fitting a $k$-additive surrogate game to randomly sampled coalition-value pairs comes with a twofold benefit. First, it reduces flexibility, leading to rapid convergence of satisfactory quality and second, the Shapley values of the $k$-additive surrogate game can be computed exactly in polynomial time. In summary, the contributions of this paper are:

   **(i)** $SVAk_{\text{ADD}}$ fits a $k$-additive surrogate game to sampled coalition values, trying to represent the underlying arbitrary value function by a simpler structure with a parameterizable degree of freedom while maintaining low representation error. The surrogate game's structure allows to compute its Shapley values in polynomial time yielding precise estimates for the original game if the representation exhibits a good fit.

  **(ii)** $SVAk_{\text{ADD}}$ does not require any structural properties of the value function. Thus, our method is domain-independent and can be applied to any cooperative game oblivious to what the players and payoffs represent. Specifically in the field of explainability, it is model-agnostic and can approximate local as well as global explanations.

 **(iii)** We empirically illustrate the utility of our method at the hand of explanation tasks. Besides demonstrating state-of-the-art approximation quality depending on the explanation type, we also shed light onto the best fitting degree of $k$-additivity.

The remainder of this paper is organized as follows. Existing works related to this paper are described in Section 2. Section 3 introduces the theoretical background behind our proposal. In Section 4, we present our novel approximation method. We conduct empirical experiments for several real-world datasets in Section 5. Finally, in Section 6, we conclude our findings and highlight directions for future works.

## 2 RELATED WORK

The problem of approximating the Shapley value, and the recent interest it attracted from various communities, lead to a multitude of diverse approaches to overcome its exponential complexity. First to mention among the class of methods that can handle arbitrary games, without further assumptions on the structure of the value function, are those which construct mean estimates via random sampling. Fittingly, the Shapley value of each player can be interpreted as the expected marginal contribution to a specific probability distribution over coalitions. Castro et al. (2009) propose with *ApproShapley* the sampling of permutations from which marginal contributions are extracted. Further works, following the paradigm of sampling marginal contributions, employ the stratification by coalition size (Maleki et al., 2013; Castro et al., 2017; van Campen et al., 2018; Okhrati & Lipani, 2020), or utilize reproducing kernel Hilbert spaces (Mitchell et al., 2022) and thus refine this approach. Departing from marginal contributions, *Stratified SVARM* (Kolpaczki et al., 2024a) splits the Shapley value into multiple means of coalition values and updates the corresponding estimates with each sampled coalition, being further refined by *Adaptive SVARM* (Kolpaczki et al., 2024b). Guided by a different representation of the Shapley value, *KernelSHAP* (Lundberg & Lee, 2017) solves an approximated weighted least squares problem, to which the Shapley value is its solution if it encompasses all coalitions. Fumagalli et al. (2023) prove its variant *Unbiased KernelSHAP* to be equivalent to a Monte Carlo technique incorporating importance sampling of single coalitions. Joining this family, Pelegrina et al. (2023a) propose $k_{ADD}$-SHAP, which consists in a local explainability strategy that formulates the surrogate model assuming a $k$-additive game[1]. The authors locally adopted the Choquet integral as the interpretable model, whose parameters have a straightforward connection with the Shapley values.

On the contrary, tailoring the approximation to a specific application of interest by leveraging structural properties, promises faster converging estimates or even closed-from polynomial solutions of the Shapley value. A prominent example is the field of data valuation (Ghorbani & Zou, 2019; Jia et al., 2019b) which assesses the significance of individual datapoints to a learning algorithm's task of producing a well-fitted model. Here, including knowledge of how datapoints tend to contribute to this task has proven to be a fruitful approach resulting in multiple tailored approximation methods Ghorbani & Zou (2019); Jia et al. (2019b;a). In similar fashion Liben-Nowell et al. (2012) proposed an algorithm leveraging supermodular cooperative games. Going one step further, by assuming the value function to be of certain parameterized shape, it is even feasible to calculate Shapley values exactly in polynomial time w.r.t. the number of involved players. Examples include the voting game (Bilbao et al., 2000) and the minimum cost spanning tree games (Granot et al., 2002) being used having found applications in operations research.

Besides the Shapley value's prominence for explaining the decision-making of a machine learning models, it has also found its way to more applied tasks. For instance, Nimmy et al. (2023) use the Shapley value to quantify each feature's impact in predicting the risk degree in managing industrial machine maintenance, Pelegrina et al. (2023b) apply it to evaluate the influence of each electrode on the quality of recovered fetal electrocardiograms, and Brusa et al. (2023) measure the features' importance towards machinery fault detection. Worth mentioning, each application requires an appropriate modelling in terms of player set and value function in order to obtain meaningful explanations. Moreover, such an analysis can be useful in feature engineering to perform feature selection. For instance, features with low relevance towards the model performance may be removed from the dataset without an impact into the quality of predictions (Pelegrina & Siraj, 2024).

## 3 THEORETICAL BACKGROUND

First, we formally introduce cooperative games and the Shapley value in Section 3.1. Next, we present in Section 3.2 the concept of $k$-additivity, constituting the core idea of our approach.

---

[1]Note that $k_{ADD}$-SHAP is limited to local explanations. In contrast, our proposed method *SVAk*$_{ADD}$ differentiates itself by its applicability to any formulation of a cooperative game. Moreover, in the context of explainable AI, it is capable of providing global explanations.

### 3.1 Cooperative games and the Shapley value

A cooperative game is formally described by $n$ players, captured by the set $N = \{1, \ldots, n\}$, and an associated payoff function $\nu : \mathcal{P}(N) \to \mathbb{R}$, where $\mathcal{P}(N)$ represents the power set of $N$. This simple but expressive formalism may for example represent a shipment coordination where companies form a coalition in order to save costs when delivering their products. In this case, the companies can be modelled as players and $\nu(A)$ represents the benefit achieved by the group of companies $A \subseteq N$. Clearly, $\nu(N)$ is the total benefit when all companies (players) form the grand coalition $N$. Commonly, one normalizes the game by defining $\nu(\emptyset) = 0$, i.e., the worth of the empty set. However, in explainability, $\nu(\emptyset)$ may take nonzero values, e.g., with no features available one may obtain a classification accuracy of 50%. In this case, one can normalize $\nu$ by simply subtracting the worth of the empty set from all game payoffs, i.e., $\nu'(A) \leftarrow \nu(A) - \nu(\emptyset)$ for all $A \subseteq N$.

A central question arising from a cooperative game is how to fairly share the worth $\nu(N)$ of the grand coalition $N$ among all participating players. The Shapley value (Shapley, 1953) emerges as the prevalent solution concept since it uniquely satisfies axioms that intuitively capture fairness (Shapley, 1953). Given the game $(N, \nu)$, the Shapley value of each player $i$ is defined as

$$\phi_i = \sum_{A \subseteq N \setminus \{i\}} \frac{(n - |A| - 1)\,|A|!}{n!} [\nu(A \cup \{i\}) - \nu(A)], \tag{1}$$

where $|A|$ represents the cardinality of coalition $A$. It can be interpreted as a player's weighted average of marginal contributions to the payoff. Among the fulfilled axioms such as null player, symmetry, and additivity (see (Young, 1985) for more details and other properties), in explainability the most useful is efficiency. It demands that the sum of all players' Shapley values is equal to the difference between $\nu(N)$ and $\nu(\emptyset)$. Mathematically, efficiency means

$$\sum_{i=1}^{n} \phi_i = \nu(N) - \nu(\emptyset). \tag{2}$$

Or, in the game theory framework where $\nu(\emptyset) = 0$, one obtains $\sum_{i=1}^{n} \phi_i = \nu(N)$. In explainability, efficiency can be used to decompose a measure of interest among the set of features. As a result, one can interpret the importance of each feature to that measure.

Unfortunately, satisfying the desired axioms in the form of the Shapley value comes at a price. According to Equation (1), the calculation requires the evaluation of all $2^n$ coalitions within the exponentially growing power set of $N$. In fact, the exact computation of the Shapley value is known to be NP-hard (Deng & Papadimitriou, 1994). Hence, its exact computation does not only become practically infeasible for growing player numbers but it is also of interest that the evaluation of only a few coalitions suffices to retrieve precise estimates. For instance, a model has to be costly re-trained and re-evaluated on a test dataset for each coalition if one is interested in the features' impact on the generalization performance. Therefore, a common goal is to approximate all Shapley values $\phi_i, \ldots, \phi_n$ of a given game $(N, \nu)$ by observing only a subset of evaluated coalitions $\mathcal{M} \subseteq \mathcal{P}(N)$. We denote the size of $\mathcal{M}$ by $T \in \mathbb{N}$ and refer to it as the available budget representing the number of samples an approximation algorithm is allowed to draw. The mean squared error (MSE) serves as a popular measure to quantify the quality of the obtained estimates $\hat{\phi}_1, \ldots, \hat{\phi}_n$ and is to be minimized:

$$\mathbb{E}\left[ \frac{1}{n} \sum_{i=1}^{n} \left( \hat{\phi}_i - \phi_i \right)^2 \right], \tag{3}$$

where the expectation is w.r.t. the (potential) randomness of the approximation strategy.

### 3.2 Interaction indices and $k$-additivity

The underlying idea of measuring the impact (or share) of a single player $i$ by means of its marginal contributions finds its natural extension to sets of players $S$ in the Shapley interaction index (Murofushi & Soneda, 1993; Grabisch, 1997a) by generalizing from marginal contributions to discrete derivatives. For any $S \subseteq N$ its Shapley interaction $I(S)$ is given by

$$I(S) = \sum_{A \subseteq N \setminus S} \frac{(n - |A| - |S|)!\,|A|!}{(n - |S| + 1)!} \left( \sum_{A' \subseteq S} (-1)^{|S| - |A'|} \nu(A \cup A') \right). \tag{4}$$

Instead of individual importance, $I(S)$ indicates the synergy between players in $S$. Although this interpretation is not straightforward for coalitions of three or more entities, it has a clear meaning for pairs. For two players $i$ and $j$, the Shapley interaction index $I_{i,j}$ quantifies how the presence of $i$ impacts the marginal contributions of $j$ and vice versa. Especially in the field of explainable AI, where players represent features, the interaction index of $S = \{i, j\}$ can be interpreted as follows:

- If $I_{i,j} < 0$, there is a negative interaction (or a redundant effect) between features $i, j$.
- If $I_{i,j} > 0$, there is a positive interaction (or a complementary effect) between features $i, j$.
- If $I_{i,j} = 0$, there is no interaction between $i, j$. Both features act independently on average.

Note that the Shapley interaction index reduces to the Shapley value for a singleton, i.e., $I(\{i\}) = \phi_i$. Moreover, there is a linear relation between the interactions and the game payoffs (Grabisch, 1997a). Indeed, from the interaction one may easily retrieve the game payoffs by the following expression:

$$\nu(A) = \sum_{B \subseteq N} \gamma_{|A \cap B|}^{|B|} I(B), \tag{5}$$

where $\gamma_{|A \cap B|}^{|B|}$ is defined by

$$\gamma_r^s = \sum_{l=0}^{r} \binom{r}{l} \eta_{s-l} \tag{6}$$

and

$$\eta_r = -\sum_{l=0}^{r-1} \frac{\eta_l}{r - l + 1} \binom{r}{l} \tag{7}$$

are the Bernoulli numbers starting with $\eta_0 = 1$.

This linear transformation recovers any coalition value $\nu(A)$ by using the Shapley interaction values of all $2^n$ coalitions, thus including the Shapley values. Therefore, $2^n$ many parameters are to be defined if the whole game is to be expressed by Shapley interactions. However, in some situations one may assume that interactions only exist for coalitions up to $k$ many players. This assumption leads to the concept known as $k$-additive games. A $k$-additive game is such that $I(S) = 0$ for all $S$ with $|S| > k$. Depending on $k$, this may significantly decrease the number of parameters to be defined. For instance, in 2-additive and 3-additive games, there are only $n(n+1)/2$, and $n(n^2+5)/6$ respectively, many interactions indices as the remaining parameters are equal to zero. Obviously, this restricts the flexibility of the game but reduces the effort when defining the unknown parameters. Indeed, for low $k$ the number of parameters increases polynomially with the number of players.

## 4   $k$-ADDITIVE APPROXIMATION APPROACH

We present in this section our proposed $SVAk_{\text{ADD}}$ approach to approximate Shapley values. It builds upon the idea of adjusting a $k$-additive surrogate game to randomly sampled and evaluated coalitions $\mathcal{M}$ (see Figure 1 for an illustration of the approach). Having fitted the surrogate game to represent the observed coalition values with minimal error, its own Shapley values can be retrieved as estimates $\hat{\phi}_1, \ldots, \hat{\phi}_n$ of the true values since the fitting promises preciseness. As the surrogate game is $k$-additive, its Shapley values can be computed exactly in polynomial time. This is due to the fact that, for $k$-additive games, $I(S) = 0$ for all $S \subseteq N$ with $|S| > k$. Therefore, by assuming $k$-additivity, the number of coalitions needed to define the whole game is reduced (as several parameters are set to zero). The drawback of this strategy is the reduction in flexibility left to model the observed game according to the obtained evaluations. However, we can still model interactions for coalitions up to $k$ players. Empirically, works in the literature (Grabisch et al., 2002; 2006; Pelegrina et al., 2020; 2023a) have been using 2-additive or even 3-additive games and the obtained results were satisfactory in modeling interactions.

Let $\mathcal{M} = \{A_1, \ldots, A_T\}$ be the set of sampled coalitions with $A_i \neq A_j$ for all $i \neq j$ and the sequence $\nu_{\mathcal{M}} = (\nu(A_1), \ldots, \nu(A_T))$ representing its evaluated coalition values. With the purpose of achieving a $k$-additive game based on the coalition evaluations $\nu_{\mathcal{M}}$, the idea in this paper consists

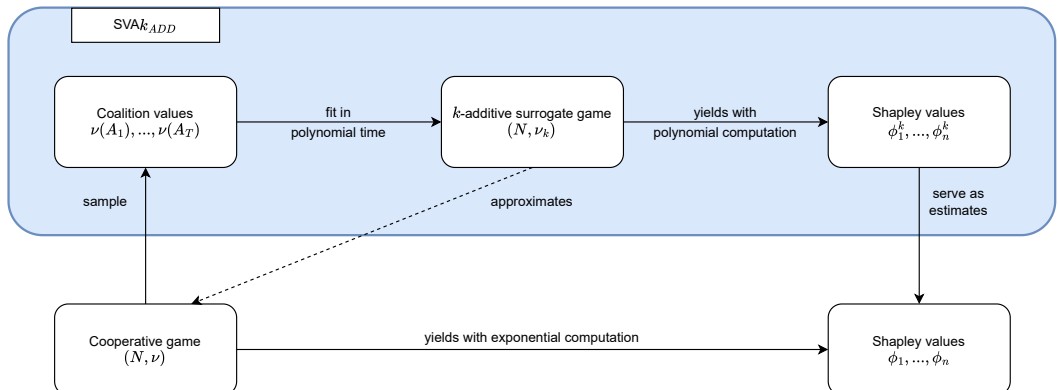

Figure 1: The from $(N, \nu)$ sampled coalition values $\nu(A_1), \ldots, \nu(A_T)$ are used to fit a $k$-additive surrogate game $(N, \nu_k)$. The Shapley values $\phi_1^k, \ldots, \phi_n^k$ of $(N, \nu_k)$ can be calculated in polynomial time by leveraging $k$-additivity. Since $\nu_k$ approximates $\nu$, these serve as estimates of the true Shapley values $\phi_1, \ldots, \phi_n$ which can only be retrieved in exponential time from $(N, \nu)$.

in retrieving a $k$-additive value function $\nu_k$ for $N$ that is as close as possible to the observations $\nu_{\mathcal{M}}$ and thus approximates $\nu$. Therefore, our goal consists in minimizing the following expression:

$$\sum_{A \in \mathcal{M}} w_A \left(\nu(A) - \nu_k(A)\right)^2 , \tag{8}$$

where $w_A$ is an importance weight associated to the coalition $A$. Recall from Equation (4) that there is a linear transformation from the value function to the interaction and Shapley values. Therefore, one may safely say that, for the $k$-additive game $\nu_k$, there exists a linear transformation

$$\nu_k(A) = \sum_{B \in \mathcal{M}} \gamma_{|A \cap B|}^{|B|} I^k(B) , \tag{9}$$

with interactions $I^k(B)$ for all $B \subseteq N$ of size $|B| \leq k$. Note that these include the Shapley values $\phi^k$ of the game $(N, \nu_k)$ since $I^k(\{i\}) = \phi_i^k$ for all $i \in N$.

As the efficiency property will explain the marginal contributions of features from the empty set to the grand coalition, it is important that our proposal can explain the difference between $\nu(\emptyset)$ and $\nu(N)$ for the true evaluations on the empty set and the grand coalition. This is ensured by imposing the following: (i) both $\emptyset$ and $N$ must be sampled and (ii) $\nu(\emptyset) = \nu_k(\emptyset)$ as well as $\nu(N) = \nu_k(N)$. For (i), one may impose in the sample strategy that such coalitions are selected with probability 1. By doing this, one ensures that $\mathcal{M} \ni \emptyset, N$. In order to satisfy (ii), one may simply include constraints ensuring that $\nu(A) = \sum_{B \in \mathcal{M}} \gamma_{|A \cap B|}^{|B|} I^k(B)$ for $A \in \emptyset, N$. With the inclusion of these elements, the resulting optimization problem that we deal with in this paper is the following:

$$\min_{I^k} \quad \sum_{A \in \mathcal{M} \setminus \{\emptyset, N\}} w_A \left(\nu(A) - \sum_{B \in \mathcal{M}} \gamma_{|A \cap B|}^{|B|} I^k(B)\right)^2$$
$$\text{s.t.} \quad \nu(\emptyset) = \sum_{B \in \mathcal{M}} \gamma_{|\emptyset \cap B|}^{|B|} I^k(B) \tag{10}$$
$$\nu(N) = \sum_{B \in \mathcal{M}} \gamma_{|N \cap B|}^{|B|} I^k(B)$$

Note that one may assign different importance degrees to the sampled coalitions. However, in our experiments, we considered the same weight for all of them (e.g., 1). We provide the analytical solution to this optimization problem in Appendix A.

A relevant aspect of our proposal is how to sample $T$ coalitions $\mathcal{M} \subseteq \mathcal{P}(N)$ in order to calculate the value functions $\nu_{\mathcal{M}}$. For this purpose, we followed the same strategy adopted in (Lundberg & Lee, 2017; Pelegrina et al., 2023a). The coalitions $A \in \mathcal{M}$ are sampled according to the probability distribution $p$ defined by

$$p_A = \frac{\pi(A)}{\sum_{B \subseteq M} \pi(B)} \quad \text{with} \quad \pi(A) = \frac{(n-1)}{\binom{n}{|A|} |A| (n - |A|)} . \tag{11}$$

---

**Algorithm 1** $SVAk_{\text{ADD}}$

---

1: **Input:** $(N, \nu), k, T$
2: $\mathcal{M} \leftarrow \{\emptyset, N\}$
3: $\nu_{\mathcal{M}} \leftarrow (\nu(\emptyset), \nu(N))$
4: $\pi(A) \leftarrow \frac{(n-1)}{\binom{n}{|A|}|A|(n-|A|)}$ for all $A \subseteq N \setminus \{\emptyset, N\}$
5: $p_A \leftarrow \frac{\pi(A)}{\sum_{B \subseteq M} \pi(B)}$ for all $A \subseteq N \setminus \{\emptyset, N\}$
6: **while** $|\mathcal{M}| < T$ **do**
7:    Sample a coalition $A \subseteq N$ with normalized distribution $p_A$ and evaluate $\nu(A)$
8:    $\mathcal{M} \leftarrow \mathcal{M} \cup \{A\}$
9:    $\nu_{\mathcal{M}} \leftarrow (\nu_{\mathcal{M}}, \nu(A))$
10:    $p_A \leftarrow 0$
11: **end while**
12: $(I^k(A))_{A \subseteq N : |A| \leq k} \leftarrow \textsc{SolveOptimization}(\mathcal{M}, \nu_{\mathcal{M}}, k)$
13: **Output:** $I^k(\{1\}), \ldots, I^k(\{n\})$

---

In order to avoid picking up the same coalition in this sampling strategy, we impose a sampling procedure without replacement. Therefore, after sampling a coalition $A$, we set $p_A$ to zero and normalize the remaining probabilities. This procedure is repeated until $|\mathcal{M}| = T$. Algorithm 1 presents a pseudo-code of our proposal. The algorithm requires the game $(N, \nu)$ (players and value function), the additivity degree $k$, and the budget $T$. Thereafter, based on the (normalized) probability distribution $p$, it samples $T$ coalitions from $\mathcal{P}(N)$ in order to define the subset $\mathcal{M}$, evaluates each, and extends $\nu_{\mathcal{M}}$. Finally, it solves the optimization problem described in Equation (10) given the importance weights $w_A$ (see Appendix A for more details). The extracted interactions $I^k(A)$ of the surrogate game also contain its true Shapley values $\phi^k$ since $I^k(\{i\}) = \phi_i^k$, which are then returned, serving as estimates $\hat{\phi}_1, \ldots, \hat{\phi}_n$ for the Shapley values $\phi$ of the considered game $(N, \nu)$.

## 5 EMPIRICAL EVALUATION

In order to assess the approximation performance of $SVAk_{\text{ADD}}$, we conduct experiments with cooperative games stemming from various explanation types. While our method is not limited to a certain domain, we find the field of explainability best to illustrate its effectiveness. We consider several real datasets as well as different tasks. The evaluation of our proposal is mainly two-fold. Not only are we interested in the comparison of $SVAk_{\text{ADD}}$ against current state-of-the-art model-agnostic methods in Section 5.2, but we also seek to investigate how the choice of the assumed degree of additivity $k$ affects the approximation quality (see Section 5.3). In the sequel of Section 5.1, we describe the utilized datasets and resulting cooperative games. For more technical details see Appendix B.

### 5.1 DATASETS

We distinguish between three explanation tasks: global feature importance, local feature attribution, and unsupervised feature importance.

Within global feature importance (Covert et al., 2020) the features' contributions to a model's generalization performance are quantified. This is done by means of accuracy for classification and the mean squared error for regression on a test set. For each evaluated coalition a random forest is retrained on a training set. We employ the *Diabetes* (regression, 10 features), *Titanic* (classification, 11 features), and *Wine* dataset (classification, 13 features).

On the contrary, local feature importance (Lundberg & Lee, 2017) measures each feature's impact on the prediction of a fixed model for a given datapoint. While the predicted value can directly be used as the worth of a feature coalition for regression, the predicted class probability is required instead of a label for classification. Rendering a feature outside of an evaluated coalition absent is performed by means of imputation that blurs the features contained information. The experiments are conducted on the *Adult* (classification, 14 features), *ImageNet* (classification, 14 features), and *IMDB* natural language sentiment (regression, 14 features) data.

In the absence of labels, unsupervised feature importance (Balestra et al., 2022) seeks to find scores without a model's predictions. This is achieved by employing the total correlation of a feature subset as its worth, since the datapoints can be seen as realizations of the joint feature value distribution. For this explanation type, we consider the *Breast cancer* (9 features), *Big Five* (12 features), and *FIFA 21* (12 features) datasets.

## 5.2 THE IMPACT OF THE ADDITIVITY DEGREE $k$

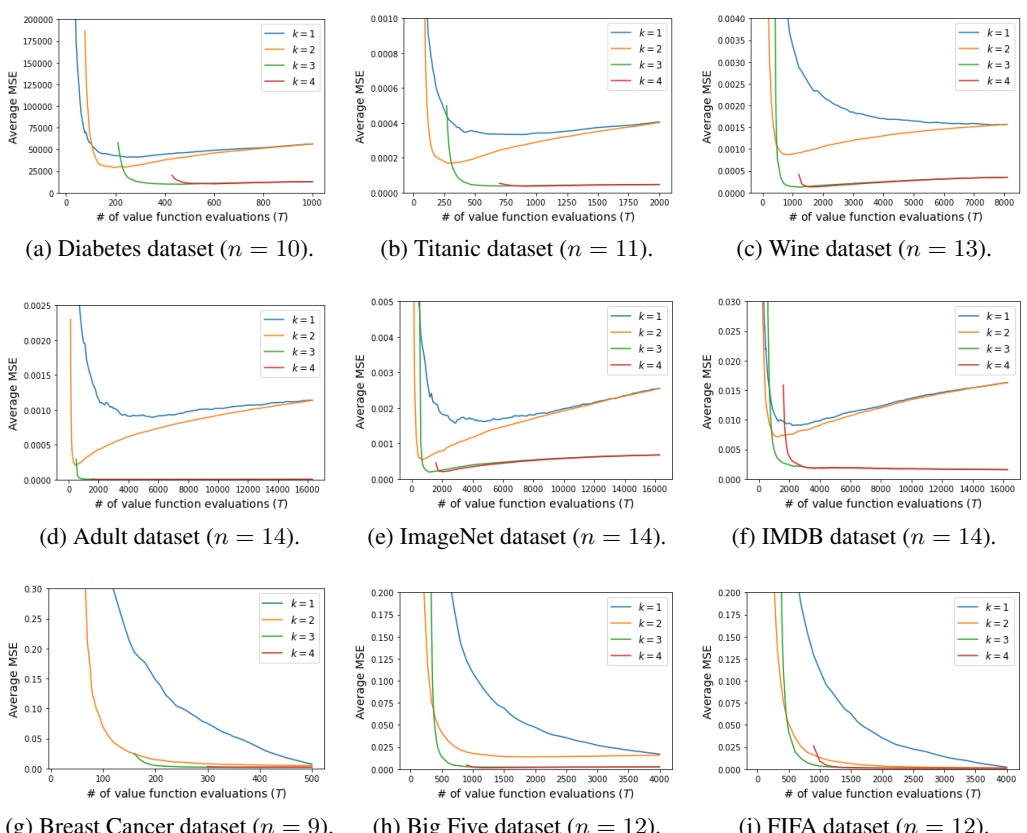

(a) Diabetes dataset ($n = 10$).     (b) Titanic dataset ($n = 11$).     (c) Wine dataset ($n = 13$).

(d) Adult dataset ($n = 14$).     (e) ImageNet dataset ($n = 14$).     (f) IMDB dataset ($n = 14$).

(g) Breast Cancer dataset ($n = 9$).     (h) Big Five dataset ($n = 12$).     (i) FIFA dataset ($n = 12$).

Figure 2: MSE of *SVAk*$_{\text{ADD}}$ averaged over 100 repetitions in dependence of available sample budget $T$ for different additivity degrees $k$. Datasets stem from various explanation types (i) global (first row), (ii) local (second row), and unsupervised (third row) with differng player numbers $n$.

In order to provide an understanding of the underlying trade-off between fast convergence (low $k$) and expressiveness (high $k$) of the surrogate game and how the crucial choice of $k$ affects the approximation quality, we evaluate *SVAk*$_{\text{ADD}}$ for different $k$. Hence, we consider different $k$-additive models, for $k \in \{1, 2, 3, 4\}$. For each dataset, $k$-additive model and different number of value function evaluations $T$, the obtained Shapley values $\phi_1^{\mathcal{M},k}, \ldots, \phi_n^{\mathcal{M},k}$ are compared with the Shapley values $\phi_1, \ldots, \phi_n$ which we calculate exhaustively in advance. We measure approximation quality of the estimates by the mean squared error (MSE) as given by Equation (3).

Figure 2 presents the obtained results for all datasets. *SVAk*$_{\text{ADD}}$ displays consistent performance curves across all datasets. Note that the curves for higher $k$ begin at points of higher budget because the greater $k$, the more coalition values are required to identify a unique $k$-additive value function that fits the observations. We explain the behavior for low $k$, specifically $k = 1$, by the model's inability to achieve a good fit due to missing flexibility. As a result, its Shapley values diverge from the true values and it reaches its optimum at relatively high MSE numbers. A similar observation can be made for the 2-additive model in both global and local tasks. It achieves good performances within a range of relatively low number of evaluations (around 500 to 1000 samples for the local

explanations with $n = 14$) but diverges as more samples are included. These findings imply that interactions up to order 2 are not sufficient to model how features jointly impact performance (global task) or prediction outcome (local task).

What is arguably unexpected is the non-monotonic behavior of some of the performance curves, in particular for $k = 2$: In some cases, the MSE decreases in the beginning and then, with additional functions evaluations, starts to increase again. Actually, one would expect that performance only improves with an increasing sample size, at least in expectation. One should note, however, that the (approximate) Shapley values are not fitted directly. Instead, they are only derived from the ($k$-additive) game that is fitted to the data, and even if the fit of this game is improved, it does not automatically imply a better fit of the Shapley values.

On the other hand, both the 3-additive and 4-additive models reach the optimum and practically remained stable as more samples are included. A slight divergence could be observed in Diabetes, Wine and ImageNet datasets, however, much lower in comparison with the 1-additive and 2-additive models. By comparing $k = 3$ and $k = 4$ variants, the choice of $k = 3$ appears preferable as it results in quicker decreasing error curves.

There is an interesting remark about the number of samples when the 3-additive model reaches the optimum. Recall that in such a model there are $n(n^2 + 5)/6$ parameters to be defined. By analyzing the obtained results, we could empirically observe that twice this value is an adequate number of value function evaluations to approximate the Shapley values (i.e., $n(n^2 + 5)/3$ sampled coalitions).

## 5.3  COMPARISON WITH EXISTING APPROXIMATION METHODS

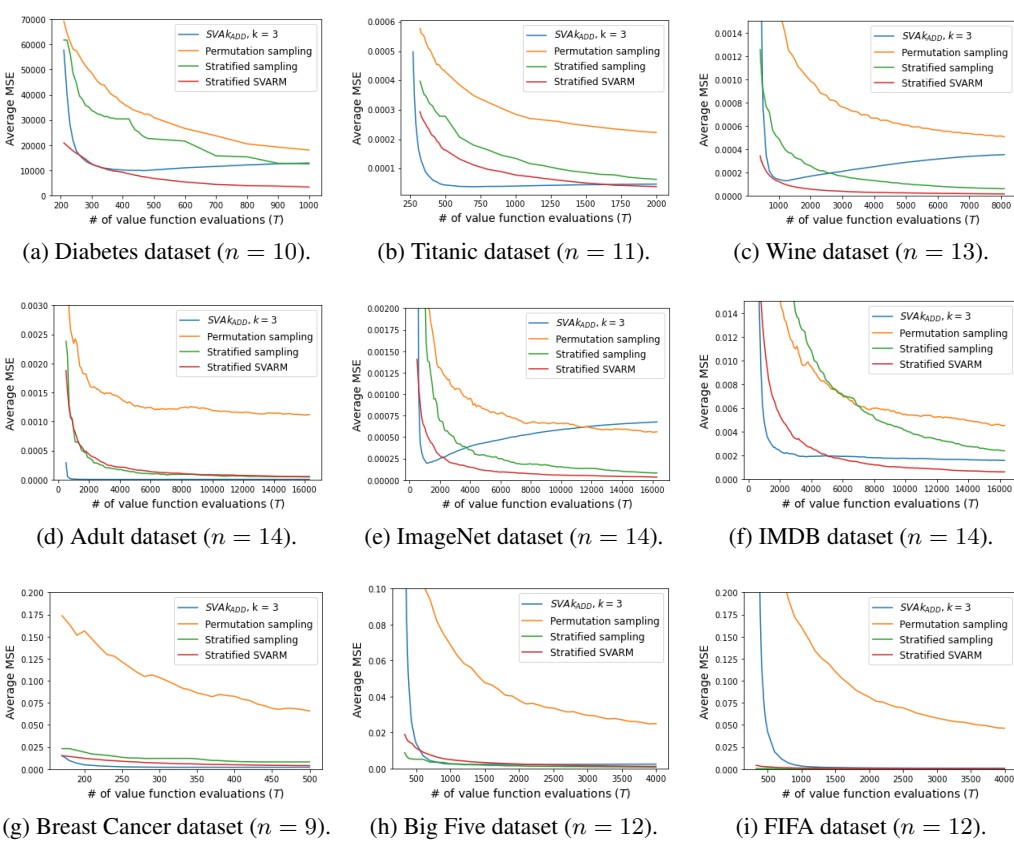

(a) Diabetes dataset ($n = 10$).  (b) Titanic dataset ($n = 11$).  (c) Wine dataset ($n = 13$).

(d) Adult dataset ($n = 14$).  (e) ImageNet dataset ($n = 14$).  (f) IMDB dataset ($n = 14$).

(g) Breast Cancer dataset ($n = 9$).  (h) Big Five dataset ($n = 12$).  (i) FIFA dataset ($n = 12$).

Figure 3: MSE of $SVAk_{\mathrm{ADD}}$ and competing methods averaged over 100 repetitions in dependence of available sample budget $T$. Datasets stem from various explanation types (i) global (first row), (ii) local (second row), and unsupervised (third row) with differng player numbers $n$.

In our second experiment, we compare $SVAk_{\text{ADD}}$ with other existing approximation methods. For instance, we consider *ApproShapley* (given here as *Permutation sampling*) Castro et al. (2009), *Stratified sampling* Maleki et al. (2013), and *Stratified SVARM* Kolpaczki et al. (2024a). For the purpose of comparison, we adopt the 3-additive model to represent $SVAk_{\text{ADD}}$ since it displays the most satisfying compromise between approximation quality and minimum required evaluations as argued in Section 5.2. Figure 3 presents the obtained results for all methods.

First to mention is that $SVAk_{\text{ADD}}$ competes consistently with Stratified SVARM for the best approxiamtion performance across most datasets. In some cases, especially, the Titanic, Adult, ImageNet, IMDB, and Breast Cancer datasets, $SVAk_{\text{ADD}}$ converges faster than its competitors. Although it remains stable, or slightly diverges with more value function evaluations, Stratified SVARM in contrast further converges to the true Shapley values, thus returning estimates of superior precision for large sample numbers. However, with the purpose of reducing the computational effort of approximating Shapley values, we argue that the performance of any approximation method within a range of low sample numbers plays an important role. Therefore, we see this advantage in $SVAk_{\text{ADD}}$, as it rapidly approximates the Shapley values with highest precision.

## 6 CONCLUSION

We proposed with $SVAk_{\text{ADD}}$ a new algorithm to approximate Shapley values. It falls into the class of approaches that fit a structured surrogate game to the observed value function instead of providing mean estimates via Monte Carlo sampling. Despite restricting the surrogate game to be $k$-additive, our developed method is model-agnostic and hence applicable to any cooperative game without posing further assumptions. We investigated empirically the trade-off that the choice of the parameter $k$ poses. Further, $SVAk_{\text{ADD}}$ exhibits a considerable reduction in estimation error for low budget ranges which indicates its suitability for use cases in which the number of players and the cost of evaluation is relatively high in comparison to the available computational resources.

**Limitations and Future Work.** While the surrogate game's flexibility increases with higher $k$-additivity, it also requires more observations to begin with in order to obtain a unique solution of the optimization problem, eventually posing a practical limit on $k$. The $k$-additive structure inherently causes a bias within the approximation as shown by our experiments, while the reduced variances of the estimates are beneficial to the approximation precision. Understanding at which budget range the inflicted bias starts to outweigh the variance reduction, indicating the point of best approximation performance, is crucial and a natural avenue for further research. We expect future investigations of differently structured surrogate games to yield likewise fruitful results and contribute to the advancement of this class of approximation algorithms.

Note that, besides the estimated Shapley values, our proposal also provides the interaction effects when $k \geq 2$. Although we did not address these parameters in this paper, future works can extract the estimated interaction indices and use them in machine learning interpretability to investigate redundant or complementary features. For instance, this could be of interest in practical applications where interaction between features are relevant as for example in disease detection.

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

## A   ANALYTICAL SOLUTION TO THE OPTIMIZATION PROBLEM

In order to solve the optimization problem presented in Equation (10), one may use a trick to remove the constraints. One may include both $\emptyset$ and $N$, as well as $\nu(\emptyset)$ and $\nu(N)$, into the objective and assign them with large weights (e.g., $w_\emptyset = w_N = 10^6$). As a consequence, one ensures that both constraints $\nu(\emptyset) = \sum_{B \in \mathcal{M}} \gamma_{|\emptyset \cap B|}^{|B|} I^k(B)$ and $\nu(N) = \sum_{B \in \mathcal{M}} \gamma_{|N \cap B|}^{|B|} I^k(B)$ are satisfied when minimizing the objective.

With the aforementioned modifications, the optimization problem can be formulated as follows:

$$\min_{I^k} \sum_{A \in \mathcal{M}} w_A \left( \nu(A) - \sum_{B \in \mathcal{M}} \gamma_{|A \cap B|}^{|B|} I^k(B) \right)^2 . \tag{12}$$

Clearly, (12) is a weighted least square problem. Indeed, assume $\mathbf{W}$ as a matrix whose diagonal elements are the weights $w_A$ for all $A \in \mathcal{M}$, $\nu_\mathcal{M}$ as the associated vector of sampled coalitions, and $\mathbf{P}$ as the transformation matrix from the generalized interaction indices to the game, i.e., $\nu_\mathcal{M} = \mathbf{P} I^k$, where $I^k = (I^k(\emptyset), \phi_1^k, \ldots, \phi_n^k, I_{1,2}^k, \ldots, I_{n-1,n}^k, \ldots, I^k(A))$, with $|A| = k$, is the vector of generalized interactions in the lexicographic order for coalitions of players such that $|A| \leq k$. In matrix notation, (12) can be formulated as

$$\min_{I^k} \left( \nu_\mathcal{M} - \mathbf{P} I^k \right)^T \mathbf{W} \left( \nu_\mathcal{M} - \mathbf{P} I^k \right) , \tag{13}$$

whose well-known solution is given by

$$I^k = \left( \mathbf{P}^T \mathbf{W} \mathbf{W} \right)^{-1} \mathbf{P}^T \mathbf{W} \nu_\mathcal{M} . \tag{14}$$

## B   COOPERATIVE GAMES DETAILS

The cooperative games used within our conducted experiments are based on explanation examples for real world data. This section complete their brief description given in Section 5. Across all cooperative games the players represent a fixed set of features given by a particular dataset.

### B.1   GLOBAL FEATURE IMPORTANCE

Seeking to quantify each feature's individual importance to a model's predictive performance, the value function is based on the model's performance of a hold out test set. This necessitates to split the dataset at hand into training and test set. Features outside of an inspected coalition $S$ are removed by retraining the model on the training set and measuring its performance on the test set. For all games we a applied train-test split of 70% to 30% and a random forest consisting of 20 trees. For classification the value function maps each coalition to the model's resulting accuracy on the test set minus the accuracy of the mode within the data such that the empty coalition has a value of zero. For regression tasks the worth of a coalition is the reduction of the model's mean squared error compared to the empty set which is given by the mean prediction. Again, the empty coalition has a value of zero.

### B.2   LOCAL FEATURE ATTRIBUTION

Instead of assessing each feature's contribution to the predictive performance, its influence on a model's prediction for a fixed datapoint can also be investigated. Hence, the value function is based on the model's predicted value.

#### B.2.1   ADULT CLASSIFICATION

A sklearn gradient-boosted tree classifies whether a person's annual salary exceeds 50,000 in the *Adult* tabular dataset containing 14 features. The predicted class probability of the true class is taken as the worth of a coalition $S$. In order to render features outside of $S$ absent, these are imputed by their mean value such that the datapoint is compatible to the model's expected feature number.

### B.2.2 Image classification

A *ResNet18* model is used to classify images from *ImageNet*. Since the for error tracking necessary exact computation of Shapley values is infeasible for the given number of pixels, 14 semantic segments are formed after applying *SLIC*. These super-pixels form the player set. Given that the model predicts class $c$ using the full image, the value function assigns to each coalition $S$ the predicted class probability of $c$ resulting from only including those super-pixels in $S$. The other super-pixels are removed by mean imputation, setting them grey.

### B.2.3 IMDB sentiment anaylsis

A *DistilBERT* transformer fine-tuned on the *IMDB* dataset predicts the sentiment of a natural language sentence between -1 and 1. The sentence is transformed into a sequence of tokens. The input sentences are restricted to sentences that result in 14 tokens being represented by players of the co-operative game. This allows to remove players in the tokenized representation of the transformer. The predicted sentiment is taken as the worth of a coalition.

### B.3 Unsupervised feature importance

In contrast to the previous settings, there is no available predictive model to investigate unlabeled data. Still, each feature's contribution to the shared information within the data can be quantified and assigned as a score. (Balestra et al., 2022) proposed to view the features $1, \ldots, n$ as random variables $X_1, \ldots, X_n$ such that the datapoints are realizations of their joint distribution. Next, the worth of a coalition $S$ is given by their total correlation

$$\nu(S) = \sum_{i \in S} H(X_i) - H(S)$$

where $H(X_i)$ denotes the Shannon entropy of $X_i$ and $H(S)$ the contained random variables joint Shannon entropy. The utilized datasets are reduced in the number of features and datapoints to ease computation. The *Breast cancer* dataset contains 9 features and 286 datapoints. The class label indicating the diagnosis is removed. From the *Big five* and *FIFA 21* dataset 12 random features are selected out of the first 50 and the datapoints are reduced to the first 10,000.

