# OpenReview forum: "Shapley Value Approximation based on k-Additive Games"
_ICLR.cc/2025/Conference — Submitted to ICLR 2025_

### Official Review · Reviewer_PQmj · 2024-11-03

**Soundness:** 3
**Presentation:** 3
**Contribution:** 2
**Rating:** 5
**Confidence:** 3

**Summary:**

This paper proposes a new algorithm SVAk_{ADD}, which leverages the Shapley information property of k-additive games to approximate the computation of Shapley values and avoid the exponential number of computation in the original version. The authors provided experiment results to show the performance of the new algorithm.

**Strengths:**

1. The efficiency improvement of Shapley value estimation is important
2. The background and preliminary are sufficiently discussed and the paper is in general easy to follow

**Weaknesses:**

1. There is no theoretical result on the estimation error bounds of the proposed algorithm
2. The computational advantage mainly works for k-additive when k=2 or 3, but probably not for higher k numbers and thus it might only have an efficiency advantage over baseline methods on particular datasets with well separated and curated features
3. The fact that the error curve is not monotonic raises questions regarding the properties of the optimization problem in SVAk_{ADD}, whereas the baseline methods all have monotonic deceasing properties.
(1). It is important to specify such condition difference if possible, e.g., convexity, potential function existence, since it might be very helpful to re-design the algorithm to satisfy such conditions and also achieve efficiency improvement
(2). Having this nike shape curve means that without knowing the ground truth, people have to use heuristics to decide the early stopping point, which is not needed in baseline methods
4. The difference between the current work and k-add SHAP is not sufficiently explained, going into the algorithm details is important since the topics are very closely related

**Questions:**

1. Under what conditions, can we conclude that increasing k can have better approximation?
2. How to evaluate a dataset to tell how close a k-additive game formulation can approximate it?

**Details Of Ethics Concerns:**

No concern for this

---

> ### Author Response · Authors · 2024-11-21
> **Answer to Reviewer PQmj**
>
> We appreciate the reviewer's effort in providing us with helpful concerns and questions and target to improve our paper along these points.
>
> Q: There is no theoretical result on the estimation error bounds of the proposed algorithm.
>
> - This is correct and we kindly refer to our answer to reviewer GBJp on the same statement.
>
> Q: The fact that the error curve is not monotonic raises questions regarding the properties of the optimization problem in $SVAk_{ADD}$, whereas the baseline methods all have monotonic deceasing properties.
>
> - We aim at fixing the non-monotonic behavior of the error curves by analytically deriving the correct weights $w_A$ such that the optimization problem in Equation (10) yields the Shapley values within its solution. For more details we kindly refer to our answer to reviewer WKUG.
>
> Q: Under what conditions, can we conclude that increasing k can have better approximation?
>
> - Setting $k=n$ recovers Equation (5) and would allow to perfectly approximate although $2^n$ many parameters need to be fitted. Thus one would assume that a representation of the cooperative game in the objective function of Equation (10) that comes closer to Equation (5) would yield a better approximation. This can be achieved by increasing $k$. Our empirical finding in Figure 2 support this conjecture.
>
> Q: How to evaluate a dataset to tell how close a k-additive game formulation can approximate it?
>
> - This question is more difficult to answer as one might think at first glance.
> One could measure the interactions of higher order than $k$ and check how close they are to zero. If they are indeed relatively close to zero then our idea of discarding these in the representation of the value function (compare Equation 5 and 10) is certainly appropriate. However, interactions are same as the Shapley value burdensome to compute and require to be approximated which brings us back to the original problem. Second, the presence of higher-order interaction does not imply the malfunctioning of a $k$-additive approach. As we conjecture, there exists a weighting $w_A$ to recover the Shapley values in the solution of the full optimization problem in Equation 10 for any $k$. Hence, it becomes only a matter of the number of sampled coalitions and not the $k$-additivity degree whether the approximation is precise or not.
>
> - This leads us to state that as of now, we do not see an approach that could answer this question in hindsight before approximation or even analyzing the value function on all coalitions.
> However, our conducted experiments suggest that a $k$-additive game can serve as a precise approximation (in comparison to other methods) for even low $k$.

---

### Official Review · Reviewer_WKUG · 2024-11-04

**Soundness:** 3
**Presentation:** 3
**Contribution:** 2
**Rating:** 3
**Confidence:** 2

**Summary:**

This paper proposes a method to approximate the Shapley value of a coalition game. The computation of the exact Shapley value is known to be a computationally hard problem for general coalition games.

The paper proposes a sampling-based algorithm that consists in evaluating the value of T subsets and then finding a k-additive game whose valuation on this T subsets is as close as possible. The approximate Shapley value can then be deduced from this set by using that when a game is k-additive, the Shapley values can be computed by a polynomial (O(n^k)) algorithm.

The evaluation of the proposed method is purely experimental and consists in showing that the method provides a "good" approximation of the Shapley value on some datasets.

**Strengths:**

The paper is relevant to the community.

The algorithm is easy to implement and seems to perform relatively well.

**Weaknesses:**

I am not a specialist of the literature on the approximation of Shapley value but I found the contributions of the paper relatively limited:

1/ I did not find a major technical novelty nor difficulty for deriving the main algorithm of the paper:
- A large part of the main ideas of the paper seem to be inspired from [Pelegrina et al 2023a] regarding the links between k-additive games and easy-to-compute Shapley values.
- Once the idea of restricting the attention to k-additive game has been given, the algorithm is relatively straighforward.

2/ The evaluation of the algorithm is limited to an empirical evaluation whose main message is not clear:
- it somehow show that the method produces results that are reasonable but they are very hard to interpret (is an MSE of 20000 good? Bad?)
- the method is compared with other heuristics (Figure 3) but it is not clear if it performs better or worse.

The empirical result could be used to provide guidance on how to tune parameters (e.g., T vs k) or why the method work in this setting and not others but this is not done.

To summarize, it seems to me that this algorithm has potential for interesting future work but that the contributions of the present paper are not good enough for ICLR.

**Questions:**

I do not have very specific questions but I am wondering if the authors would have an explanation for why the performance of the algorithm is not monotone in T (in Figure 2). This is especially visible for small k but seems also true for larger values of T. The authors acknowledge that this is unexpected but is there some explanation?

Also, there seems to be an argument that using T=n(n^2+5)/3 sampled coalitions is good. Could this be proven? Could this be studied empirically? Is this true for larger / smaller models (i.e. with very different n).

Finally, the studied model are relatively small (for these models, the Shapley values can be computed in close form). Is there room for larger experiments? For instance for models for which the Shapley value could be computed in close form? Would this method be comparable with other existing methods?

---

> ### Author Response · Authors · 2024-11-21
> **Answer to Reviewer WKUG**
>
> We thank the reviewer for the raised questions and plan to clarify them in the revised version of our paper.
>
> Q: I am wondering if the authors would have an explanation for why the performance of the algorithm is not monotone in T (in Figure 2). This is especially visible for small k but seems also true for larger values of T. The authors acknowledge that this is unexpected but is there some explanation?
>
> - We are confident to say that this caused by the choice of the weights $w_A$ in Equation (10). We suspect that there exists a unique choice that yields the Shapley value as the solution of the optimization problem if all coalitions have been sampled, similar to KernelSHAP (Lundberg \& Lee, 2017) which has its won unique weights proven by Charnes et al. (1988): "Extremal Principle Solutions of Games in Characteristic Function Form: Core, Chebychev and Shapley Value Generalizations". In: Sengupta, J.K., Kadekodi, G.K. (eds) Econometrics of Planning and Efficiency. Advanced Studies in Theoretical and Applied Econometrics, vol 11. Springer, Dordrecht.
>
> - This implies, that any other choice of $w_A$ causes a bias as the solution of the full optimization problem is not the Shapley value. The analytical derivation of these weights remains open and would contribute significantly to the efficacy of our proposed method as well as provide further theoretical foundation.
>
> Q: Also, there seems to be an argument that using $T=n(n^2+5)/3$ sampled coalitions is good. Could this be proven? Could this be studied empirically? Is this true for larger / smaller models (i.e. with very different n).
>
> - We heuristically derived this value as it is the twice the number of parameters fitted for the case of $k=3$. We have no proof for this but our empirical results indicate this to be a sensible choice as the error curves reach its minima close to it, although we do not provide further studies to particularly answer this question. We refrain from further optimization of this number, falling into overfitting, as this is a fairly simple choice.
>
> Q: Finally, the studied model are relatively small (for these models, the Shapley values can be computed in close form). Is there room for larger experiments? For instance for models for which the Shapley value could be computed in close form? Would this method be comparable with other existing methods?
>
> - The Shapley value can be computed for any model, player number, or value function in general in closed form as given by Equation (1). We chose datasets with up to $n=14$ many features because we are limited in computational resources to compute the true Shapley values which are used to calculate the MSE within our experiments. For larger feature numbers we would not be able to track the approximation error.
>
> - There exists more restricted models, like linear models for example, for which the exact calculation of the Shapley values is feasible in polynomial time. While this would allow us to conduct studies with higher $n$, we question the meaningfulness since such models are of lesser concern regarding explainability or at least approximation.

---

> > ### Comment · Reviewer_WKUG · 2024-11-26
> >
> > I thank the authors for their answer, which I think is convincing.
> >
> > Out of curiosity: regarding the error for large model. I understand that you cannot go beyond n=14 because you do not have a "ground truth" for larger models. Yet, you claim that your method is designed for larger n.  How can you guarantee that the error will not explode as n grows? Did you study the error as a function of n? (theoretically or experimentally).

---

### Official Review · Reviewer_GBJp · 2024-11-07

**Soundness:** 3
**Presentation:** 2
**Contribution:** 1
**Rating:** 3
**Confidence:** 3

**Summary:**

This paper proposes approximately computing Shapley values using limited (k) size of the interaction set in Shapley interaction index (on top of sampling of coalition). The authors evaluate their results in some standard datasets.

**Strengths:**

- The approach is simple and intuitive
- The approach is needed for the importance task for XAI.

**Weaknesses:**

This paper has many weaknesses. The most critical weakness is that there is no approximation guarantee, even as the paper presents an approximation algorithm. In game theory literature, abstractions of game (as done in this paper) are not very useful unless they come with some approximation bounds. Plus, I did a google search and found work that also restrict interaction to smaller sets such as https://proceedings.mlr.press/v206/bordt23a/bordt23a.pdf and other work on Shapley-Taylor index - how does this work compare to these works?

The experiments show only MSE results - how about showing the actual XAI result with feature importance? In fact, the experiments section starts by mentioning all these XAI datasets and global/local feature importance - but no results showing the feature importance?

Lastly, the writing is lacking in many places, to the extent that it makes the paper difficult to read. For example, takes these lines from the paper : "Hence, its exact computation does not only become practically infeasible for growing player numbers but it is also of interest that the evaluation of only a few coalitions suffices to retrieve precise estimates. For instance, a model has to be costly re-trained and re-evaluated on a test dataset for each coalition if one is interested in the features’ impact on the generalization performance." - here coalitions are not evaluated but used in the evaluation of Shapley values. The second sentence has grammatical errors.
- In Eq 11, what is M in denominator of p_A (this is just M, not calligraphic M)
- Algorithm 1 is written where it appears as if p_A is computed for all subsets A - that itself is exponential. The sampling must be written appropriately.

**Questions:**

Please respond to weaknesses.

---

> ### Author Response · Authors · 2024-11-21
> **Answer to Reviewer GBJp**
>
> We are thankful for the reviewer's raised concerns and attention to detail that give us the opportunity to further improve our work. We will incorporate the mentioned points together with our explanations into the revised version of our paper.
>
> Q: The most critical weakness is that there is no approximation guarantee, even as the paper presents an approximation algorithm. In game theory literature, abstractions of game (as done in this paper) are not very useful unless they come with some approximation bounds.
>
> - While some approximation methods that view the Shapley value as an expected value come with approximation guarantees, this is not the case for the class of approaches that are based on weighted least squares optimization. In fact and to the best of our knowledge, there exist no approximation guarantees for such methods.
>
> - We would be very thankful if the reviewer could provide us literature in case we are wrong and have misse dthese results. In particular, there exists no guarantee for the popular KernelSHAP (Lundberg \& Lee, 2017) and the difficulty to obtain one is further elaborated by Covert \& Lee (2021): "Improving kernelshap: Practical shapley value estimation using linear regression" (AISTATS). It remains as one of the key challenges in this field.
>
> Q: Plus, I did a google search and found work that also restrict interaction to smaller sets such as https://proceedings.mlr.press/v206/bordt23a/bordt23a.pdf and other work on Shapley-Taylor index - how does this work compare to these works?
>
> - Bordt \& Luxburg (2023) tackle the question of how to decompose an observed effect fully only among interactions up to some order $k$ by exploiting the connection of Shapley values and interactions to generalized additive models, and propose, as a consequence, a new interaction index. Their empirical results motivate our conceptual idea to discard interactions of higher order, higher than $k$ in our case, since these are close to zero, from Equation (5) which simplifies our optimization problem in Equation (10).
>
> Q: The experiments show only MSE results - how about showing the actual XAI result with feature importance? In fact, the experiments section starts by mentioning all these XAI datasets and global/local feature importance - but no results showing the feature importance?
>
> - The choice to not show feature importance values but the resulting approximation quality has multiple reasons. First, the MSE is central to judge the quality of approximation methods but the underlying Shapley values, used as feature importance scores, give no clue on the precision of these methods which is why they play a less important role.
>
> - Second, as pointed out by us in the introduction, our proposed method is independent of the actual cooperative game and thus not limited to approximating feature importance. Moreover, it can even be further applied outside the realm of explainabitlity or machine learning in learning. To keep this context-agnostic spirit and speak to a wider audience, we refrain from showing feature importance scores.
>
> Q: For example, takes these lines from the paper: "Hence, its exact computation does not only become practically infeasible for growing player numbers but it is also of interest that the evaluation of only a few coalitions suffices to retrieve precise estimates. For instance, a model has to be costly re-trained and re-evaluated on a test dataset for each coalition if one is interested in the features’ impact on the generalization performance." - here coalitions are not evaluated but used in the evaluation of Shapley values. The second sentence has grammatical errors.
>
> - With all respect, we can not find a grammatical error in the second instance. In all politeness, we would like to ask the reviewer to precisely point it out such that we can improve readability.
>
> Q: Algorithm 1 is written where it appears as if $p_A$ is computed for all subsets $A$ - that itself is exponential. The sampling must be written appropriately.
>
> - We agree, the pseudocode is simplified to hide implementation details and ease understanding. For a practical solution we refer to our answer to reviewer kEpp.

---

> > ### Comment · Reviewer_GBJp · 2024-11-24
> > **More comments**
> >
> > Thanks for the response. I am not a cooperative game person, but my comment on approximation is a general comment for results that the author claim is a general game theory result beyond quantifying feature importance. Being a theory result, I am surprised by the lack of any approximation guarantee.
> >
> > Also, I am not sure why feature importance is not useful. The MSE by meaning the mean hides where the error is concentrated. A visual representation of feature importance with and without approximation can also shed light on the approximation, meaning answering questions such as do the small remaining errors make the most important feature appear unimportant?
> >
> > I checked the grammar by AI tools and AI finds it to be fine, but here is what I think is the problem: costly is used as an adjective and an adjective must describe a noun but train (or trained) is used as a verb. "costly training" is fine since training is a noun but "costly train" does not sound right to me. Anyway, this is a minor point as this is not an English paper.

---

> ### Author Response · Authors · 2024-11-25
> **Response to Reviewer GBJp**
>
> Thanks for the engagement!
>
> Regarding the approximation guarantee, there exists theoretical results implying gurantees for the approximation of the Shapley value but for a different class of algorithms. Approximation methods that consider the Shapley value as some sort of expected value of a distribution (for example a distribution of marginal contributions $\Delta_i(A)$) and leverage that by estimating that expectation through sampling come indeed with approxiamtion guarantees, promiment examples are ApproShapley (Castro et al. 2009), Stratified Sampling (Maleki et al., 2013), and Stratified SVARM (Kolpaczki et al. 2024). However, obtaining such results is by far more diffcult (and so far not done) for weighted least squares approaches like ours. The work by Covert & Lee (2021) describes that in more detail.
>
> We did not mean to say that feature importance is not useful. In fact, we motivate our paper also by relying on its recent popularity. It is just that we see the MSE as a sufficient measure to judge approximation quality as done by many others. We agree that one could investigate further by comparing the rankings of players obtained by sorting by (estimated) Shapley values.
>
> It is correct that adjectives are used to describe nouns but in this case here "costly" is and adverb standing in front of the verb "retraining". We appreciate the reviewer's effort to clarify this!

---

### Official Review · Reviewer_TUGh · 2024-11-07

**Soundness:** 2
**Presentation:** 2
**Contribution:** 1
**Rating:** 3
**Confidence:** 2

**Summary:**

This paper proposes a fast heuristic for computing Shapley value. The idea is to restrict to k-additive games (i.e., just assume this structure applies), which leads to a polynomial time algorithm for computing the Shapley value. The overall algorithm uses limited number of coalitions to derive a k-additive game that best fits the studied game, and then derive the Shapley value of the k-additive game, which serves as a fast heuristic for estimating the true Shapley values.

**Strengths:**

The k-additive assumption seems to be a reasonable assumption for a lot of the settings, so it is likely that the proposed heuristic would perform well in those settings.

**Weaknesses:**

There are a few downsides of the proposed approach. First, it is not clear how to pick k. It seems fairly arbitrary to choose k=3. Second, the performance of the proposed approach does not consistently outperform the existing baselines. It appears that the baseline algorithms have the advantage that with more and more samples, they converge to accurate Shapley values, but the proposed algorithm does not have this property. Third, other than experiments, there is no discussion on when the k-additive game assumption (for the most part) holds and how to test for it.

**Questions:**

I see that the number of function evaluations is sometimes a few hundred and sometimes 16k in Figure 3.

Why the different numbers? Why these numbers?

Is there any practical setting where we can only afford to sample a few hundred times?

---

> ### Author Response · Authors · 2024-11-21
> **Answer to Reviewer TUGh**
>
> We thank the reviewer for raising these questions. In order to clarify our paper, we will first respond to the raised questions and thereafter provide a general discussion about our proposal and the obtained results.
>
> Q. I see that the number of function evaluations is sometimes a few hundred and sometimes 16k in Figure 3. Why the different numbers? Why these numbers?
>
> When calculating the exact Shapley values (see Eq. (1)), one needs all possible function evaluations. This number depends on the number of features. Indeed, there are $2^n$ function evaluations, where $n$ is the number of features. In Figure 3, we evaluated the strategies along with the number of function evaluations available to estimate the Shapley values. Therefore this number varies from a few hundred when the number of features is low (e.g., for $n=9$, we have $2^9=512$ function evaluations) to thousands when the number of features is high (e.g., for $n=14$, we have $2^{14}=16384$ function evaluations).
>
> Q. Is there any practical setting where we can only afford to sample a few hundred times?
>
> The idea in assuming $k$-additivity is to reduce the number of function evaluations to accurately estimate the Shapley values. Therefore, even with a high number of features (e.g., for Adult dataset where there are $n=14$ features), a reduced number of function evaluations can be used to estimate the Shapley values.
>
> It is important to remark that when assuming $k$-additivity, we will not be able to model the whole game, as one restricts the interactions for coalitions of at most $k$ players (or features).
>
> It is known from the literature that interactions among players (see the references provided in the first paragraph of Sec. 4) up to $k=2$ or $k=3$ bring satisfactory results in modeling interactions. Indeed, the coalition values for high $k$ frequently boil down to be additive. These findings motivate our approach, as we can reduce the number of needed function evaluations to estimate the Shapley values. Surely, as can be seen in Figure 3, our proposal based on $k=3$ diverges from the exact Shapley values when more functions evaluations are available. However, it rapidly achieves the minimum for a few number of feature evaluations. In such scenarios, we see a perspective to conduct further investigations verifying the impact of interactions greater than $k$ in our estimates.

---

### Official Review · Reviewer_kEpp · 2024-11-08

**Soundness:** 2
**Presentation:** 3
**Contribution:** 3
**Rating:** 5
**Confidence:** 3

**Summary:**

This paper considers the problem of approximating the Shapley value in polynomial time. To do so the authors propose a novel approach of approximating the value by defining a surrogate $k$-additive game.They propose an algorithm that finds the Shapley values for a $k$-additive game in polynomial time and a method of sampling that helps us use this algorithm to approximate the Shapley value in the general case. They provide empirical evaluations to support this approach. Figure 1 presents a good summary of the whole contribution of this work.

**Strengths:**

The paper is well-written and the problem is very well-motivated and natural. The idea of using a $k$-additive surrogate game also seems novel.

**Weaknesses:**

- It is not clear what the running time of the proposed algorithm is. You are presenting an algorithm to approximately solve a problem which we can exactly solve in exponential time, but it is not really clear what the running time is. How much is this better than exponential? I don't think that this algorithm is polynomial in terms of $k$ (which is fine) but you need to be more formal than just saying the algorithm runs in polynomial time. The least that I expect is to see the running time of the algorithm for different values of $k$ in the experiments.
- It is not clear how you sample $T$ coalitions in polynomial time. If you first find the probability of each coalition, then since we have exponentially many coalitions the running time is not polynomial. I think further details are needed here.
- I think it is necessary to mention what tool/mechanism you use to solve the optimisation problem. In addition it is better to clearly state how many variables and constraints you have in the optimisation.
- The plots are too small and hard to read. You should keep a few of them in the body and differ the rest to the appendix.

**Questions:**

Please explain how you sample a coalition in polynomial time, and how does the running time of Algorithm 1 depend on different parameters.

---

> ### Author Response · Authors · 2024-11-21
> **Answer to Reviewer kEpp**
>
> We thank the reviewer for the suggestions. In the revised version, we will evaluate to move some plots to appendix and increase the size of the remaining ones in the main text.
>
> With respect to the optimization problem, we described in Appendix A how we can analytically solved it. In this formulation, we do not have constraints and the number of variables depended on the assumed $k$-additivity. Indeed, we have $\sum_{i=1}^k {n \choose k}$ variables, where $n$ is the number of features. We can clarify this aspect in the revised version.
>
> Q. Please explain how you sample a coalition in polynomial time, and how does the running time of Algorithm 1 depend on different parameters.
>
> Our pseudocode for Algorithm 1 is simplified to ease understanding of the approximation procedure and thus it does not present the sampling of a coalition in a practical way as it hides details. Indeed, assigning a probability to each coalition individually within the powerset causes exponential effort. Instead, one can exploit that the probabilities are equal for coalitions of the same size. This allows to group the coalitions within buckets of the same size and hence only $n-1$ many buckets need to be assigned with a probability value. The sampling is now divided in two steps. First a bucket is chosen, and next a coalition from that buckets is sampled uniformly at random.
>
> The runtime can be divided into the time need for sampling and that for solving the approximated optimization problem. As we usually need several computations for sampling the game payoff, solving a single quadratic optimization problem will not impact the computational time.

---

### Official Review · Reviewer_Dd1R · 2024-11-11

**Soundness:** 3
**Presentation:** 2
**Contribution:** 2
**Rating:** 5
**Confidence:** 4

**Summary:**

This paper provides a novel method to approximate the Shapley value. While no theoretical approximation guarantees are provided in this paper, an empirical evaluation is provided in the context of determining the contribution of different features to model accuracy / MSE in classification / regression problems through their Shapley values, i.e., the marginal contribution to model performance due to the features. The empirical approximation ratio of Shapley values obtained by different approximation algorithms are compared by computing the MSE with the true, exact Shapley values. The main takeaway the authors highlight is that the newly proposed method achieves a competitive MSE to other algorithms.

The main technical contribution is a novel approach for approximating the Shapley values of players in a cooperative game where a valuation function assigns a value to each coalition. Their approach involves:
- Sampling T coalitions and computing their exact Shapley interaction values exactly.
- Then, a surrogate k-additive cooperative game, and corresponding valuation function, is "learned" using existing (to the Authors: please correct me if I am wrong here) techniques which provides an analytical solution which can be performed in polynomial time in a way that minimizes the MSE of (weighted) Shapley interaction values. Here, a k-additive game is one where the interaction values of coalitions of size greater than k is 0.
- Then, Shapley values of players in the k-additive game is known to be computable in polynomial time which will be used as estimates of the Shapley values of players in the original game.

In the empirical evaluations, the features in a machine learning problem are considered as the players in a cooperative game, and the goal is to determine or "explain" the contribution of each feature towards the performance of some learning algorithm.

**Strengths:**

- The technique of using k-additive games to approximate the Shapley values is novel and may be of interest beyond the use case highlighted by the authors. The paper makes a clever use of existing results. I find the problem of approximating Shapley values and the application of explaining feature importances to be well motivated.
- The method is well motivated and the different components seem sound and worth evaluating.
- The paper is overall well organized and it is easy to follow the high level ideas.
- The experimental results validate their approach and the proposed algorithms achieve a competitive approximation ratio with other algorithms under different "budget constraints" on the number of sampled coalitions.

**Weaknesses:**

- A running time comparison of the different algorithms would have provided a useful way to compare against different approximation algorithms which seem to achieve a very similar approximation ratio.
- It is counterintuitive that the MSE of Shapley value approximations gets worse as the number of samples increase. There is a brief intuition provided in the paper, but it is concerning. As Figure 3 shows, for certain problems, competing algorithms obtain better approximations as number of samples increases, but the proposed method gets worse. It is not clear from the paper if this comes with some other benefit e.g. running time. Due to this, I am not confident about the significance of the new algorithm. There does not seem to be a clear takeaway for when a practitioner should use this algorithm.
- It is not clear what the real world importance of using a small number of samples is, which is the regime in which the proposed method outperforms others. This will likely need further experiments.

**Questions:**

- Could you address the main question of what the takeaway from this paper is? E.g. is there a clear running time or other resource usage advantage of practical relevance using the proposed method? Could you please share any experimental results?
- Could you elaborate on the motivation for focusing on the Shapley values of features, as opposed to e.g. the training data points? Was this evaluated?
- One of the claimed advantages is that the proposed method does not make any assumptions on the value functions. Could you elaborate on the practical importance of this? For example, are there examples of practically relevant games where the value functions result in poor approximations by other methods that are handled better by the proposed method?
- Could you elaborate on the downstream effect of better estimates of Shapley values on model performance on the machine learning task? I do not find any experiments on this. Maybe I am missing something, or could you discuss their exclusion?

---

> ### Author Response · Authors · 2024-11-21
> **Answer to Reviewer Dd1R**
>
> First of all, we appreciate the reviewer's comments and the careful reading of this paper. In the sequel, we provide the answers for the raised questions.
>
> Answer to Q1.
>
> It is known from the literature that the Shapley value has been used to extract interpretability from black-box models. However, it has a clear drawback as the computation needed increases exponentially with the number of features. This highlights the importance of adopting approximation strategies to estimate Shapley values as accurately as possible with a lower number of sampled evaluations. Therefore, the main takeaway from this paper is an approach that can be used to rapidly approximate Shapley values (in polynomial time). Although in this paper we considered datasets with at most 12 features (in order to be able to calculate the exact Shapley values based on all feature evaluations), this approximation is of importance especially in scenarios with a large number of features, which makes the exact calculation infeasible.
>
> With respect to computational time, note that each function evaluation requires extracting the game payoff. In global machine learning interpretability, this means retraining the model and calculating the performance for each coalition. Practically, all approximation methods adopt this strategy with a subset of evaluations. In our proposed approach, we used a strategy which consists in solving an unconstrained quadratic optimization problem. Therefore, the running time of such a solution is much lower in comparison to calculating  hundreds (or even thousands) of function evaluations. We can further discuss this aspect and present computational time results in the revised version of this paper.
>
> Answer to Q2.
>
> Surely, focusing on explaining data points has its interest in understanding the local result achieved by a particular instance. However, aiming at generalizing how features contribute towards the model performance, focusing on single points may lead to biased results. And the global understanding of features contributions is suitable to provide insights into the model’s overall behavior. This information is useful to generalize the explanation of achieved decisions by assuming multiple inputs, not just specific instances. Moreover, the knowledge of global feature importance may help practitioners to conduct feature engineering, as features with no (or even negative) contribution may be removed from the model without decreasing the model's performance. Also, when sensitive features are assigned high impact on the model performance, this may indicate a source of unfair results. In this scenario, the practitioner may adopt strategies to mitigate disparate outcomes provided by the use of such features.
>
> Answer to Q3.
>
> Indeed, our proposal can be applied to any task formulated on the basis of a cooperative game. For instance, local explanations are generally limited to machine learning tasks. Our approach may be used in machine learning tasks as well as in classical cooperative game scenarios. Although our experiments focus on machine learning explainability, works in the literature address the approximation of Shapley values in other situations, such as the glove game, the airport game and the voting game (see Benati, S., López-Blázquez, F., Puerto, J. (2019). A stochastic approach to approximate values in cooperative games. European Journal of Operational Research, 279(1), 93-106). Therefore, we see that our approach could also be useful for such problems.
>
> Answer to Q4.
>
> Better estimates of Shapley values can help practitioners in several ways. The most straightforward benefit is the improvement of interpretability on how features impact the model performance. This is of interest, for example, in healthcare, as the analysis of information collected from patients (e.g., symptoms and/or medical examinations) may help physicians to detect a disease. Another advantage of a better estimation consists in improving feature selection. Features with no relevance into the model performance may be removed from the dataset in order to simplify the model. If the estimated Shapley values are of low precision, one may wrongly conduct feature selection. We also highlight the benefit of an accurate estimation when analyzing bias towards sensitive groups. If the Shapley values indicate that sensitive features (e.g., the ones that brings information associated to gender, race, religion, etc.) have high impact on the model performance, one should be aware of possible disparate outcomes. Therefore, the accurate approximation will help in adopting calibration strategies to mitigate unfair results.

---

### Meta-Review · Area_Chair_t8C6 · 2024-12-17

**Metareview:**

Reviewers liked the novel idea proposed in the paper (that uses k-additive games to approximately compute the Shapley value). However, critical concerns were raised on the guarantee of the proposed approach, e.g., lack of theoretical analysis, unclear when k-additive games are a good guarantee, how to choose k, and lack of more informative empirical studies. Unfortunately these concerns were not addressed in the rebuttal. We hope the authors find the reviews helpful. Thanks for submitting to ICLR!

**Additional Comments On Reviewer Discussion:**

See above

---

### Decision · Program_Chairs · 2025-01-22

Reject